# High Sensitivity C-Reactive Protein Increases the Risk of Carotid Plaque Instability in Male Dyslipidemic Patients

**DOI:** 10.3390/diagnostics11112117

**Published:** 2021-11-15

**Authors:** Manuel Scimeca, Manuela Montanaro, Marina Cardellini, Rita Bonfiglio, Lucia Anemona, Nicoletta Urbano, Elena Bonanno, Rossella Menghini, Viviana Casagrande, Eugenio Martelli, Francesca Servadei, Erica Giacobbi, Arnaldo Ippoliti, Roberto Bei, Vittorio Manzari, Massimo Federici, Orazio Schillaci, Alessandro Mauriello

**Affiliations:** 1Department of Experimental Medicine, University of Rome “Tor Vergata”, 00133 Rome, Italy; manuela.montanaro@uniroma2.it (M.M.); rita.bonfiglio@uniroma2.it (R.B.); anemona@uniroma2.it (L.A.); elena.bonanno@uniroma2.it (E.B.); francescaservadei@gmail.com (F.S.); erica.giacobbi@gmail.com (E.G.); alessandro.mauriello@uniroma2.it (A.M.); 2San Raffaele University, Via di Val Cannuta 247, 00166 Rome, Italy; 3Faculty of Medicine, Saint Camillus International University of Health Sciences, Via di Sant’Alessandro 8, 00131 Rome, Italy; 4Department of Systems Medicine, University of Rome Tor Vergata, 00133 Rome, Italy; cardellini@med.uniroma2.it (M.C.); menghini@med.uniroma2.it (R.M.); viviana.casagrande@uniroma2.it (V.C.); federicm@uniroma2.it (M.F.); 5Center for Atherosclerosis, Policlinico Tor Vergata, 00133 Rome, Italy; 6Nuclear Medicine Unit, Department of Oncohaematology, Policlinico “Tor Vergata”, Viale Oxford 81, 00133 Rome, Italy; n.urbano@virgilio.it; 7Department of General and Specialist Surgery “P. Stefanini”, Sapienza University of Rome, 00185 Rome, Italy; eugenio.martelli@aorncaserta.it; 8Division of Vascular Surgery, S. Anna and S. Sebastiano Hospital, 81100 Caserta, Italy; 9Vascular Surgery Unit, Department of Biomedicine and Prevention, University of Rome Tor Vergata, 00133 Rome, Italy; ippoliti@med.uniroma2.it; 10Department of Clinical Sciences and Translational Medicine, University of Rome “Tor Vergata”, 00133 Rome, Italy; bei@med.uniroma2.it (R.B.); manzari@uniroma2.it (V.M.); 11Department of Biomedicine and Prevention, University of Rome “Tor Vergata”, Via Montpellier 1, 00133 Rome, Italy; orazio.schillaci@uniroma2.it; 12IRCCS Neuromed, Via Atinense, 18, 8607 Pozzilli, Italy

**Keywords:** atherosclerosis, hs-CRP, carotid plaque, cardiovascular risk factors, dyslipidemic patients

## Abstract

Background: The aim of this study was to evaluate how the high sensitivity C-reactive protein (hs-CRP) values influence the risk of carotid plaque instability in association with other cardiovascular risk factors. Methods: One hundred and fifty-six carotid plaques from both symptomatic and asymptomatic patients requiring surgical carotid endarterectomy were retrospectively collected. According to the modified American Heart Association, atherosclerosis plaques have been histologically distinguished into unstable and stable. The following anamnestic and hematochemical data were also considered: age, gender, hypertension, diabetes mellitus, smoking habit, therapy, low-density lipoprotein (LDL)-C, kidney failure and hs-CRP. Results: The results of our study clearly show that high levels of hs-CRP significantly increase the carotid plaque instability in dyslipidemic patients. Specifically, a 67% increase of the risk of carotid plaque instability was observed in patients with high LDL-C. Therefore, the highest risk was observed in male dyslipidemic patients 2333 (95% CI 0.73–7.48) and in aged female patients 2713 (95% CI 0.14–53.27). Discussion: These data strongly suggest a biological relationship between the hs-CRP values and the alteration of lipidic metabolism mostly in male patients affected by carotid atherosclerosis. The measurement of hs-CRP might be useful as a potential screening tool in the prevention of atheroscletotic disease.

## 1. Introduction

Atherosclerosis is a pathological condition characterized by a chronic active inflammatory condition of the major arteries leading to the development of intimal plaques [1,2,3,4]. Atherosclerosis begins at a young age and slowly progress for decades, without necessarily becoming symptomatically evident. The clinical symptoms of atherosclerosis occur in adulthood and usually involve a destabilization of the plaque with intimal rupture and thrombosis [5,6,7,8].

Among clinical manifestations of atherosclerosis, the ischemic stroke correlated with extracranial carotid plaques, remains an important cause of adult mortality and disability worldwide despite the efforts pointing to the prevention of risk factors such as hypercholesterolemia, diabetes and hypertension [1,2,3,4]. Currently, cardiovascular and cerebrovascular diseases represent the leading cause of death worldwide. With regard to the atherosclerosis of carotids, a recent meta-analysis and modelling study [9] analyzed data in people aged 30–79 and found that the prevalence of carotid plaque was about 816 million cases, and that of carotid stenosis was about 58 million cases in 2020. These data highlight the impact of carotid atherosclerosis on human life and put the emphasis on the need to improve our understanding about the mechanisms explaining the development of cardiovascular adverse events.

It is known that the low-grade inflammation, as well as the alteration of lipid metabolism, are two of the main causes of carotid plaque destabilization [10,11]. In addition, several investigations highlighted a connection between systemic inflammation and the alteration of lipid metabolism [11,12,13].

In this scenario, a current challenge is the identification of more sensitive and specific biomarkers capable of stratifying patients according to the risk of carotid plaque instability before clinical syndromes develop.

In our previous studies, we demonstrated that different types of plaque calcifications, whether or not combined with some risk factors, may increase the risk of plaque rupture and thrombosis [13,14,15,16,17,18]. It should also be emphasized that in the last years, several circulating molecules have been proposed as prognostic biomarkers of atheromatic plaque instability. Among these, numerous authors focused their investigations on the C-reactive protein (CRP), a circulating protein whose levels indicate the body’s systemic inflammatory state. It has been proposed that the increase of CRP significantly increases the risk of acute cardiovascular disease related to the rupture of coronary plaques [19]. The hs-CRP test is considered more sensitive than a standard CRP test. A high hs-CRP value is today associated with a 10% increase of having a cardiovascular event within the 10 years following the test [19]. However, conflicting data are reported about the carotid district [20,21] and no definitive data have been released about the possible role of hs-CRP in the management of patients affected by carotid atherosclerosis. Hence, it is possible to hypothesize that hs-CRP may also increase the plaque rupture risk in synergy with the simultaneous presence of other cardiovascular risk factors.

Starting from these considerations, the aim of this study was to evaluate how the hs-CRP values influence the risk of carotid plaque instability in association with other cardiovascular risk factors such as age, gender, hypertension, diabetes, smoking habit, low HDL and high LDL-C.

To this end, 156 carotid plaques, with associated anamnestic and hematochemicals data, have been retrospectively collected.

## 2. Materials and Methods

### 2.1. Cases Selection and Histology

In this study, 156 carotid plaques from both symptomatic (major stroke or transient ischemic attack—TIA) and asymptomatic patients who underwent carotid endarterectomy (CEA) were retrospectively collected at the University of Tor Vergata (Rome, Italy) from 2016 to date. The sampling collection and analysis methods have been previously reported [6,13]. Briefly, samples were fixed in 10% buffered formalin, quickly decalcified (Surgipath Decalcifer II, LEICA, Western Road, Stratford Upon Avon, UK), cut transversely every 5 mm, embedded in paraffin and stained with haematoxylin-eosin for the morphological analysis. Only intact carotid plaques, from patients with a complete clinical and laboratory assessment of the major cardiovascular risk factors, were included in the study. Regarding the exclusion criteria, only a clinical history of infection, inflammatory diseases, arthritis and autoimmunity excluded patients from the study.

Informed consent was obtained from each patient. The study protocol was approved by the Ethics Committee of our institution (reference no. 129.18, 26 July 2018).

According to the modified American Heart Association (AHA), atherosclerosis plaques have been histologically distinguished into unstable and stable [22]. Unstable plaques consisted of: (a) thrombotic plaque associated with rupture or erosion of the cap; (b) healed plaque with a thrombus in organization; and (c) vulnerable plaque or thin-cap fibro-atheroma (TCFA) characterized by a fibrous cap less than 165 µm thick, heavily infiltrated by macrophages CD68 positive (>25 per high magnification field), without plaque rupture. The other plaques, classified as stable, included: (a) fibroatheromata, such as plaques with a large lipidic necrotic core and thick non-inflamed cap; (b) fibrocalcific plaques with large calcification without extensive inflammation; and (c) fibrous plaques mainly constituted by fibrous tissue.

A histopathologic examination was performed, by two blinded pathologists (A.M., F.S.), according to the definitions reported above. The inter-observer reliability was >98%.

### 2.2. Risk Factors Definition

Clinical records were reviewed for all cases to determine the risk factors profile.

The presence of hypertension was assessed if patients showed systolic BP > 140 mmHg and/or a diastolic blood pressure > 90 mmHg with or without antihypertensive treatment at the time of carotid endarterectomy.

The other major risk factors were considered as follows: diabetes mellitus, patients with fasting blood glucose > 126 mg/dL and/or following oral treatment or insulin therapy; patients with tobacco dependence were categorized as smokers, if the consumption was more than 10 cigarettes/day, while those who had stopped smoking for >5 years were considered as non-smokers.

In order to evaluate the levels of atherogenic cholesterol, low-density lipoprotein cholesterol (LDL-C) was calculated by the Friedewald equation [23] as follows: LDL-C = cholesterol − (HDL-C + (triglycerides/5)). An LDL-C value of >100 mg/dL was used as the cut-off between high and low levels.

In addition, the estimated glomerular filtration rate (eGFR) was defined and calculated using the CKD-EPI (chronic kidney disease epidemiology collaboration) equation. Values of Egfr < 60 mL/min per 1.73 m^2^, present for >3 months, were considered as the cut-off for renal chronic disease (GFR categories: G3a-G5) [24].

Serum levels of high-sensitivity C-reactive protein were measured by using an immunoturbidimetric method (Abbott Diagnostics, Milan, Italy). Patients were subdivided in two groups by using the 50th percentile of hs-CRP (i.e., the value of 0.26 mg/L) as the cut-off value.

### 2.3. Statistical Analysis

Data were analyzed using SPSS version 16.0 (SPSS Inc, Chicago, Illinois, USA) software. Continuous variables were expressed as the mean (SD)or ± SE. The Shapiro–Wilk test was used to statistically assess the normal distribution of the data. Comparisons between continuous variables were performed using the independent Student t-test or the Wilcoxon rank sum test. Categorical data were analyzed using the chi-square test or the Fisher exact test.

A multivariate analysis using stepwise logistic regression (using the “enter” method for variable selection) was utilized to identify independent risk factors which significantly correlate with the presence of plaque destabilization. The odds ratio of an unstable plaque for the different risk factors was evaluated by logistic regression using the value of EXP (B), where B represents the logistic coefficient. Moreover, to evaluate the possible impact of hs-CRP blood levels on the instability of the carotid plaques, the odds ratio of each risk factor in association with high hr-CRP (≥0.26 mg/L) was also calculated by using logistic regression.

For all analyses a 2-tailed *p* value < 0.05 was considered statistically significant.

## 3. Results

### 3.1. Baseline Data

Baseline data of the patients are reported in Table 1.

The mean age of 156 patients at the time of surgical CEA was 72.3 + 8.31 years (range 43–91), 113 (72.4%) were male (72.6 + 8.09 years; range 43–91) and 43 (27.6%) were female (71.6 + 8.91 years; range 47–86). Sixty-six (38.7%) patients were symptomatic (affected by ipsilateral major stroke or TIA), while 90 (61.3%) were asymptomatic who underwent CEA for high grade carotid stenosis (>60%), assessed by echography or, in rare cases, by bilateral computed tomography (CT) angiography.

All enrolled patients presented at least one risk factor. Among single risk factors, the hypertension was the most frequently observed (96 patients, 61.5%).

In 71 out of 156 patients (45.5%) unstable plaques were found. They consisted of 42 thrombotic plaques (26.9%) (all from symptomatic patients) associated with the rupture of a thin fibrous cap rich in inflammatory cells, 13 TCFA (eight from symptomatic and five from asymptomatic patients) and 12 plaques with an organized acute thrombus (all symptomatic) characterized by a network of large, thin-walled vascular channels and a variable number of macrophagic cells loaded with hemosiderin within the area of an acute thrombus. In the remaining four unstable plaques, a calcified nodule with a protrusion covered by an extremely thin fibrous cap was found into the lumen. The remaining 85 carotids (54.5% of cases) showed a stable plaque, characterized by a variable lipid-necrotic core containing extracellular lipid, cholesterol crystals and necrotic debris covered by a thick fibrous cap with few inflammatory cells. In 33 plaques (both stable and unstable) a large calcification was observed.

### 3.2. Plaque Instability and Risk Factors

As demonstrated by the multivariate analysis, the presence of unstable carotid plaques was not correlated with the finding of specific risk factors (see Table 2). Only a significant correlation with the gender was observed, as males showed a higher incidence of unstable plaques than females (*p* = 0.008). In fact, unstable plaques were observed in 56 of the 113 (49.55%) male patients who underwent CEA and only in 15 of the 43 (34.8%) female patients.

The greatest odds ratio for an unstable plaque was observed in patients with high-LDL (1.27, 95% CI 0.67–2.43). The other risk factors showed a lower odds ratio for an unstable plaque: 1.03 (0.54–1.98) for patients with hypertension, 0.64 (95% CI 0.34–1.22) for diabetes, 0.87 (95% CI 0.41–1.86) for smoking habit, 1.24 (95% CI 0.63–2.45) for patients with low-HDL and 1.01 (95% CI 0.50–2.04) for kidney failure.

### 3.3. Impact of hs-CRP on the Risk of Plaque Instability

Concerning the case selection investigated here, the hs-CRP does not represent an independent risk factor for carotid plaque instability if not associated with other risk factors (odds ratio 1.03 (95% CI 0.75–1.41)). Conversely, high hs-CRP values (>0.26 mg/dL) significantly increase the carotid plaque instability when associated with specific risk factors (Table 2). Specifically, our data showed a great increase in the risk of carotid plaque instability for dyslipidemic patients with high LDL-C. The odds ratio for patients with high LDL-C increased to 1.86 (95% CI 0.68–5.12) with an increase of 67% (Table 2). In terms of percentages, the high level of hs-CRP also increases the risk of plaque instability for patients affected by diabetes (61%) and hypertension (10%) even though the absolute odds ratio value does not reach relevant thresholds (diabetes 1.04 (95% CI 0.39–2.71); hypertension 1.14 (95% CI 0.41–3.16)).

Furthermore, the separate analysis of male and female patients (Table 3) showed that high hs-CRP values significantly increase the risk of carotid plaque instability when associated with high LDL-C only in male patients (odds ratio 2333 (95% CI 0.75–1.41)). Conversely, the risk of carotid plaque instability increases in female patients with age (age ≥ 70).

## 4. Discussion

The results of our study clearly show that high levels of hs-CRP significantly increase the carotid plaques’ instability mostly in male dyslipidemic patients. Therefore, these data strongly suggest a biological relationship between the hs-CRP values and the alteration of lipidic metabolism in male patients affected by carotid atherosclerosis.

Although some of the circulating molecules lack specificity because they are involved in systemic and local physio-pathological processes, many of them have been proposed as acute phase biomarkers associated with the progression of carotid plaques [25]. Among these, the hs-CRP has been widely investigated as an acute-phase marker of inflammation whose value is associated with the risk of developing cardiovascular events. Indeed, hs-CRP is able to reflect low but persistent levels of inflammation.

Recently, a large series of prospective studies involving about 1600 patients with asymptomatic carotid atherosclerosis (average follow-up 11.81 years), demonstrated that patients with high levels of hs-CRP (>0.29 mg/dL) were characterized by a significant increase in cardiovascular mortality [26]. However, an unequivocal association between hs-CRP values and the carotid atherosclerosis has not been identified yet. Indeed, whilst some investigations indicate that high blood hs-CRP levels are able to predict the presence of atheromatic carotid plaque [20,21,27], numerous other studies neither found this association [28,29] nor any relationship with the degree of carotid stenosis [21,30]. These apparent incongruences could be explained by the influence of other local and/or systemic inflammatory conditions such as infection, arthritis and autoimmunity on the hs-CRP levels. However, an hs-CRP value of 2.6/mg/mL in male patients with an alteration of the lipid metabolism significantly increases the risk of carotid plaque instability, regardless of these confounding factors. It should also be noted that patients with a history of inflammatory disease were excluded in our study. Concerning female patients, a high hs-CRP does not increase the risk of carotid plaque instability related to the high LDL. Conversely, high hs-CRP seems to be associated with an increase of plaque instability in aged female patients. However, it is important to note that the analysis of risk factor odd ratios in the female population showed a large confidence interval due to the low number of analyzed samples (*n* = 44). In particular, the association between high hs-CRP and age 70 displayed the larger confidence interval. Therefore, this datum needs to be confirmed in large cohort studies. Confidence intervals significantly closer to the odd ratio values were observed in the male population. It is known that the systemic chronic inflammation can influence the lipidic metabolism by altering the pathways involved in synthesis and transport of cholesterol [31]. Specifically, several inflammatory cytokines modulate the cholesterol synthesis inducing a decrease in HDL-C, as well as the impairment in cholesterol transport. In addition, a persistent systemic inflammation can negatively influence the metabolism of apolipoproteins, enzymes, antioxidant capacity and ATP binding cassette A1-dependent efflux [32,33]. The effects of inflammation on cholesterol metabolism generally stimulate compensatory changes, such as the synthesis and accumulation of phospholipid-rich VLDL (very low density-lipoprotein), which trigger a vicious circle that enhances both phenomena (inflammation and alteration of lipidic metabolism), thus resulting in a hypertriglyceridemia condition [32]. The final step of these events is a complete alteration in the cholesterol metabolism with the consequent increase in blood level of LDL-C and decrease in HDL. Hence, the increase in the risk of plaque instability in male dyslipidemic patients with high hs-CRP can be explained by the alterations of cholesterol metabolism related to the chronic inflammation.

From a clinical point of view, data here reported allow us to reconsider the prognostic value of hs-CRP. In fact, despite the fact that the blood values of this molecule can fluctuate in response to several local and systemic conditions, high hs-CRP could identify patients with a very high risk of developing cardiovascular (CV) events, if associated with a dyslipidemic condition [19,20,21]. In particular, male patients with low HDL and high LDL would be particularly at risk of plaque rupture when the hs-CRP blood level exceeds 0.26 mg/L (50th percentile).

It is important to note that high hs-CRP does not significantly increase the risk of plaque instability for other investigated CV risk factors, such as hypertension, diabetes, smoke habit and the presence/absence of plaque calcifications. An interesting observation is the lack of association between the presence of plaque calcifications and hs-CRP blood level. Indeed, our and other studies demonstrated that the plaque calcification is an active process that can be modulated by the inflammation [16,18,34,35,36,37], as occur in the physiological bone mineralization [38,39]. This evidence suggests that the calcification of the carotid plaque could be related to specific inflammatory cytokines [25,37] rather than to a specific low-grade inflammatory state.

All together the data shown here could deeply change the management of male patients affected by carotid atherosclerosis since the dyslipidemia includes numerous biochemical disorders associated with adverse CV events. Abundant evidence displays that lifetime cumulative exposure to lipids strongly impacts the risk of CV events associated with the rupture of carotid plaques [40]. According to this study, hs-CRP levels may further refine high LDL-C risks in the setting of primary prevention. Male subjects with elevated levels of both high LDL-C and hs-CRP appear to be more exposed to the risk of carotid plaque instability.

Thus, early identification of atherogenic dyslipidemias, especially in male subjects with high hs-CRP blood levels (a very large subset of patients), may influence medical choices in order to guide primary prevention. The measurement of systemic inflammation by serum biomarkers’ evaluation, such as hs-CRP, could help to guide clinical decisions, although this would need to be confirmed in larger cohort studies/trials designed to specifically answer this question.

## 5. Conclusions

In this study, the great impact of high hs-CRP blood level on carotid plaque instability in dyslipidemic patients was identified. Therefore, our findings identify hs-CRP determination in association with high LDL-C as a potential screening tool with prognostic/therapeutic implications, further emphasizing the role of both cholesterol metabolism and inflammation in CV disease.

## Figures and Tables

**Table 1 diagnostics-11-02117-t001:** Baseline characteristics of patients.

	*N* (%) or Mean (SD)
**Total**	*N* = 156
**Age**	72.3 (8.6)
**Gender**	
Male	113 (72.4%)
Female	43 (27.6%)
**Cerebrovascular disease**	
Symptomatic patients	66 (38.7%)
Asymptomatic patients	90 (61.3%)
**Risk factors**	
Hypertension [22]	96 (61.5%)
Diabetes [22]	73 (46.8%)
Smoking habit [22]	35 (22.4%)
High LDL-C [23]	61 (39.1%)
Kidney failure [24]	45 (30.2%)
**Drugs**	
Statins	110 (70.5%)
Antihypertensive drugs	92 (59.0%)
**Histological type of carotid plaque**	
Stable plaques	85 (54.5%)
Fibroatheromata	24 (15.4%)
Fibrocalcific	61 (84.6%)
Unstable plaques	71 (45.5%)
Thrombotic plaque	42 (26.9%)
With a thrombus in organization	12 (7.7%)
TCFA	13 (8.3%)
Calcified nodule	4 (2.6%)

**Table 2 diagnostics-11-02117-t002:** Plaque instability and risk factors.

	Stable Plaques (85 Cases)	Unstable Plaques (71 Cases)	Odds Ratio without High hs-CRP (95% CI)	Odds Ratio with hs-CRP (95% CI)	Δ (%)
Age 70 (years + SD)	53 (62.4%)	46 (64.8%)	1.11 (0.58–2.14)	1.00 (0.38–2.64)	−10%
Gender					
Male	57 (67.1%)	56 (78.9%)	0.55 (0.26–1.13)	0.25 (0.72–0.85)	−55%
Female	28 (32.9%)	15 (21.1%)			
Hypertension	52 (61.2%)	44 (62.0%)	1.03 (0.54–1.98)	1.14 (0.41–3.16)	+10%
Diabetes	44 (51.8%)	29 (40.8%)	0.64 (0.34–1.22)	1.04 (0.39–2.71)	+61%
Smoking habit	20 (23.5%)	15 (21.1%)	0.87 (0.41–1.86)	0.70 (0.23–2.16)	−21%
High LDL-C	31 (36.5%)	30 (42.3%)	1.27 (0.67–2.43)	1.86 (0.68–5.12)	+67%
Kidney failure	25 (30.1%)	20 (30.3%)	1.01 (0.50–2.04)	1.25 (0.43–3.64)	−21%

**Table 3 diagnostics-11-02117-t003:** Impact of high hs-CRP in male and female patients.

	Male (113 Cases)	Female (43 Cases)
Age 70 (years + SD)	0.849 (0.29–2.50)	2.713 (0.14–53.27)
Hypertension	0.914 (0.27–3.06)	1.577 (0.23–10.78)
Diabetes	1.239 (0.41–3.70)	0.461 (0.58–2.14)
Smoking habit	0.759 (0.22–2.56)	0.000 (0.00)
High LDL-C	2.333 (0.73–7.48)	0.272 (0.15–4.94)
Kidney failure	1.368 (0.39–4.70)	0.551 (0.23–13.15)

## Data Availability

Data will be provided on request.

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
