# Peer review of "High Sensitivity C-Reactive Protein Increases the Risk of Carotid Plaque Instability in Male Dyslipidemic Patients"

_diagnostics, 2021, doi:10.3390/diagnostics11112117_

Round 1

Reviewer 1 Report

The odds ratio of all results shows a wide range from 0 through 1- to larger values. It is a wide range where the actual value can be both below and above 1. It is not clear whether, it is a benefit or a risk.

Author Response

Ref.: Manuscript: diagnostics-1413147

"High sensitivity C-Reactive protein increases the risk of carotid plaque instability in male dyslipidemic patients "

Submitted to: Diagnostics Journal

Before we begin the point by point review of the list of concerns, we would like to thank the Reviewer for their comments on how to improve the manuscript, which has been revised accordingly, as well as the Editors for calling for a new submission of an improved version of our manuscript.

Reviewer#1

The odds ratio of all results shows a wide range from 0 through 1- to larger values. It is a wide range where the actual value can be both below and above 1. It is not clear whether, it is a benefit or a risk.

Reply: we would like to thank the Reviewer for reviewing our work.

As concern the Reviewer#1 comment, there are several odds ratio values ≥ 1 such as those related to Hypertension, High LDL-C, and Kidney failure.

In addition, according to the suggestions of the reviewer#3, in the new version of the manuscript we reported a separate description of the results for men and women (see table 3).

This new data further highlighted the increase risk of plaque instability in male dyslipidemic patients with high hs-CRP; Odds ratio 2,333 (95% CI 0.73-7.48)

Reviewer 2 Report

The authors focused on a study of the High sensitivity C-Reactive protein increases the risk of carotid plaque instability in dyslipidemic patients. This is an interesting and comprehensive study. The article is well structured. Two tables in the text are very clearly written.

In my opinion:

  • The abstract presents an accurate description of this study.
  • An Authors was conducted adequate literature review.
  • The references support the rationale for reporting the study.
  • The patients are described adequately.
  • The management of the study is effectively described.
  • Valid and reliable outcome measures are utilized.
  • The conclusions are appropriate.

Overall impression about the quality of the study  is good.

Key points to consider:

In the paper sometimes the words “smoking”, “kidney”, “table”, “low” was written with capital letter in other cases with lowercase letter please correct with one of the versions.

Line 36 – explain the abbreviation LDL-C - first time in article

Line 39 – “ca-rotid” please fix spelling “carotid”

Line 40 – “pa-tients” please fix spelling “patients”

Line 41 – explain the abbreviation HDL - first time in article

Line 81 – “The high-sensitivity C-reactive protein (hs-CPR) 81 test is…” please fix to “The hs-CPR test is…”

Line 124 – explain the abbreviation BP

Line 138 – 1.73 m2 ‘2’ should be superscript, please correct in whole manuscript

Line 145 – “...as the mean ± SD....” please fix to “…as the mean (SD)…”

Line 156 – “hr-CPR” please fix spelling

Line 167 – explain the abbreviation CT - first time in article

Line 241 – explain the abbreviation VLDL - first time in article

Line 255 – explain the abbreviation CV - first time in article

Author Response

Ref.: Manuscript: diagnostics-1413147

"High sensitivity C-Reactive protein increases the risk of carotid plaque instability in male dyslipidemic patients "

Submitted to: Diagnostics Journal

Before we begin the point by point review of the list of concerns, we would like to thank the Reviewer for their comments on how to improve the manuscript, which has been revised accordingly, as well as the Editors for calling for a new submission of an improved version of our manuscript.

Reviewer#2

The authors focused on a study of the High sensitivity C-Reactive protein increases the risk of carotid plaque instability in dyslipidemic patients. This is an interesting and comprehensive study. The article is well structured. Two tables in the text are very clearly written.

In my opinion:

The abstract presents an accurate description of this study.

An Authors was conducted adequate literature review.

The references support the rationale for reporting the study.

The patients are described adequately.

The management of the study is effectively described.

Valid and reliable outcome measures are utilized.

The conclusions are appropriate.

Overall impression about the quality of the study is good.

Reply: we would like to thank the Reviewer for their comments on how to improve the manuscript, which has been revised accordingly.

Key points to consider:

In the paper sometimes the words “smoking”, “kidney”, “table”, “low” was written with capital letter in other cases with lowercase letter please correct with one of the versions.

Reply: thanks for this point out. We corrected the text accordingly.

Line 36 – explain the abbreviation LDL-C - first time in article

Line 39 – “ca-rotid” please fix spelling “carotid”

Line 40 – “pa-tients” please fix spelling “patients”

Line 41 – explain the abbreviation HDL - first time in article

Line 81 – “The high-sensitivity C-reactive protein (hs-CPR) 81 test is…” please fix to “The hs-CPR test is…”

Line 124 – explain the abbreviation BP

Line 138 – 1.73 m2 ‘2’ should be superscript, please correct in whole manuscript

Line 145 – “...as the mean ± SD....” please fix to “…as the mean (SD)…”

Line 156 – “hr-CPR” please fix spelling

Line 167 – explain the abbreviation CT - first time in article

Line 241 – explain the abbreviation VLDL - first time in article

Line 255 – explain the abbreviation CV - first time in article

Reply: we corrected all the mistakes highlighted by the reviewer.

Reviewer 3 Report

The article is devoted to the study of the actual problem of destabilization of atherosclerotic plaques and the factors affecting this process.

I have important comments that should improve the article.

  1. It is necessary to improve the description of research methods. It is necessary to indicate the age limits of patients, both for men and women. There are 2 times more men than women. It is necessary to provide references to recommendations according to the criteria of which all risk factors were assessed (hypertension, high LDL-C, glucose, etc.). It is necessary to indicate which test was determined by the ELISA system for CRP.
  2. It is necessary to improve the description of the results, to make this section wider. It is necessary to make an additional table with a separate description of the results for men and women. Perhaps the authors will get the differences between men and women regarding the effect of CRP on the instability of atherosclerotic plaques of the carotid arteries.
  3. It is necessary to additionally highlight and enhance the novelty of the results obtained. Indeed, a huge amount of research in the world is devoted to the study of CRP.
  4. In the list of references it is necessary to indicate DOI.
  5. It is necessary for the text of the article to be additionally reviewed by a professional translator of the English language. 

Author Response

Ref.: Manuscript: diagnostics-1413147

"High sensitivity C-Reactive protein increases the risk of carotid plaque instability in male dyslipidemic patients "

Submitted to: Diagnostics Journal

Before we begin the point by point review of the list of concerns, we would like to thank the Reviewer for their comments on how to improve the manuscript, which has been revised accordingly, as well as the Editors for calling for a new submission of an improved version of our manuscript.

Reviewer#3

The article is devoted to the study of the actual problem of destabilization of atherosclerotic plaques and the factors affecting this process.

Reply: we would like to thank the Reviewer for their comments on how to improve the manuscript, which has been revised accordingly.

I have important comments that should improve the article. It is necessary to improve the description of research methods.

It is necessary to indicate the age limits of patients, both for men and women.

Reply: In the method section we reported the age limits of patients, both for men and women.

The mean age of 156 patients at time of surgical CEA was 72.3 + 8.31 years (range 43- 91), 113 (72.4%) were male (72.6 + 8.09 years; range 43-91) and 43 (27.6%) were female (71.6 + 8.91 years; range 47-86).

There are 2 times more men than women.

Reply: thanks for this point out. In the new version of our manuscript, we provided a separate description of the results for men and women. In this context, we discussed the different number of cases among the groups. According to this, we emphasized the effects of high hs-CPR in male dyslipidemic patients rather than in female patients.

It is necessary to provide references to recommendations according to the criteria of which all risk factors were assessed (hypertension, high LDL-C, glucose, etc.).

Reply: In the table 2 we added the reference for each risk factor assessed in this study.

It is necessary to indicate which test was determined by the ELISA system for CRP.

Reply: we added the following sentence in the new version of the manuscript

Method section

Serum levels of high-sensitivity C-reactive protein were measured by using an immuno-turbidimetric method (Abbott Diagnostics, Milan, Italy).

It is necessary to improve the description of the results, to make this section wider. It is necessary to make an additional table with a separate description of the results for men and women. Perhaps the authors will get the differences between men and women regarding the effect of CRP on the instability of atherosclerotic plaques of the carotid arteries.

Reply: thanks for this important suggestion.

We added a new table (table 3) in which the impact of high hs-CRP on carotid plaque instability, in association with other risk factors, were analyzed in male and female patients separately.

Table 3. Impact of high hs-CRP in male and female patients

Male

(113 cases)

Female

(43 cases)

Age 70 (yrs + SD)

Hypertension

Diabetes

Smoking habit

High LDL-C

Kidney failure

0,849 (0.29-2.50)

0,914 (0.27-3.06)

1,239 (0.41-3.70)

0,759 (0.22-2.56)

2,333 (0.73-7.48)

1,368 (0.39-4.70)

2,713 (0.14-53.27)

1,577 (0.23-10.78)

0,461 (0.58-2.14)

0,000 (0.00-)

0,272 (0.15-4.94)

0,551 (0.23-13.15)

Data of this new analysis are very interesting.

Indeed, we found that high hs-CRP values significantly increase the carotid plaque instability when associated with high LDL-C only in male patients.

Text was modified as follow:

Noteworthy, the separate analysis of male and female patients showed that high hs-CRP values significantly increase the risk of carotid plaque instability when associated with high LDL-C only in male patients (odds ratio 2,333 (95% CI 0.75-1.41)). Conversely, the risk of carotid plaque instability increases in female patients with the age (age ≥70).

Moreover, we also changed the title of the study

High sensitivity C-Reactive protein increases the risk of carotid plaque instability in male dyslipidemic patients

Lastly, we modified the abstract e discussion according to the discoveries here described.

It is necessary to additionally highlight and enhance the novelty of the results obtained. Indeed, a huge amount of research in the world is devoted to the study of CRP.

Reply: thanks for this point out. In the discussion section, we added some sentences that further highlight and enhance the novelty of our results.

In the list of references it is necessary to indicate DOI.

Reply: done

It is necessary for the text of the article to be additionally reviewed by a professional translator of the English language.

Reply: done

Round 2

Reviewer 3 Report

l have not comment

Author Response

We would like to thank the reviewer#3 for giving us the opportunity to improve our paper